# Nutritional status, environmental enteric dysfunction, and prevalence of rotavirus diarrhoea among children in Zambia

**Aybüke Koyuncu**[1]*, **Michelo Simuyandi**[1], **Samuel Bosomprah**[1,2], **Roma Chilengi**[1]

**1** Centre for Infectious Diseases Research in Zambia, Lusaka, Zambia, **2** Department of Biostatistics, School of Public Health, University of Ghana, Accra, Ghana

\* akoyuncu@berkeley.edu

## Abstract

### Background

Rotavirus is the most common cause of fatal diarrhoeal disease among children under the age of five globally and is responsible for millions of hospitalizations each year. Although nutritional status and environmental enteric dysfunction (EED) are recognized as important predictors of susceptibility to diarrhoeal disease, no research to date has examined the mechanisms by which undernutrition and EED may protect against prevalence of rotavirus infection.

### Methods

We utilized data collected from a study evaluating the effectiveness of Rotarix™ vaccine against severe gastroenteritis among children under the age of 5 in Zambia. The prevalence of malnutrition, wasting, and stunting at the time of study enrollment was calculated using WHO child growth standards. Commercial ELISA kits were used to assess levels of faecal biomarkers for EED: alpha-1-antitrypsin and myeloperoxidase, and calprotectin. Separate multivariate logistic regression models were used to examine each measure of nutritional status and rotavirus diarrhoea including and excluding adjustment for EED.

### Results

In models that did not include adjustment for EED, malnourished children had 0.66 times the odds of having rotavirus diarrhoea compared to children with normal nutritional status (95% CI: 0.42, 1.0; p = 0.07). EED severity score was significantly higher among controls asymptomatic for diarrhoeal disease compared to cases with rotavirus diarrhoea (p = 0.02).

### Conclusion

The morphological changes associated with EED may confer protection against rotavirus infection and subsequent diarrhoeal disease among children. Further research is critically needed to better understand the complex mechanisms by which nutritional status and EED may impact susceptibility to rotavirus in early life.

**Data Availability Statement:** The underlying data set cannot be made publicly available because it contains human research participant data; however, it can be made available to any interested

researchers upon request. The Centre for Infectious Disease Research in Zambia (CIDRZ) Ethics and Compliance Committee is responsible for approving such requests. To request data access, one must write to the Secretary to the Committee/Head of Research Operations, Ms. Hope Mwanyungwi (contact via Hope. Mwanyungwi@cidrz.org) and the corresponding author (Aybüke Koyuncu) mentioning the intended use for the data, contact information, a research project title, and a description of the analysis being proposed as well as the format it is expected. The requested data should only be used for the purposes related to the original research or study. The CIDRZ Ethics and Compliance Committee will normally review all data requests within 48–72 hours (Monday-Friday), and provide notification if access has been granted or additional project information is needed.

**Funding:** This work was undertaken with support from the Bill and Melinda Gates Foundation (https://www.gatesfoundation.org/) grant number OPP1033211. There was no additional external funding received for this study. The funder had no role in study design, data collection and analysis, decision to publish, or preparation of the manuscript.

**Competing interests:** The authors have declared that no competing interests exist.

## Background

There are approximately 1.7 billion cases of diarrhoea globally each year, representing the second leading cause of death globally among children under five [1]. In Zambia, diarrhoeal disease resulted in the death of approximately 3,000 children under the age of 5 in 2016 [1, 2]. Rotavirus remains the most common cause of moderate-to-severe diarrhoea globally, with over 50% of deaths attributable to rotavirus occurring in Sub-Saharan Africa [3].

Malnourished children are disproportionately affected by morbidity and mortality attributable to diarrhoea due to the widely recognized detrimental impact of undernutrition on immune function [4–6]. Counter to abundant evidence demonstrating the ways in which undernutrition can impair the body's immune system and result in higher incidences of common childhood illnesses [4–6], numerous observational studies have recently suggested an association between "better" nutritional status and increased susceptibility to rotavirus infection early in life [7–9]. For example, in a longitudinal birth cohort in Bangladesh, malnourished children had 0.57 times the odds of rotavirus isolation from their stool compared to well-nourished children [8]. Evidence suggestive of a protective association between malnutrition and rotavirus diarrhoea has also been found in Zambia, where rotavirus infection was less common in hospitalized children who were malnourished compared to those with normal nutritional status [7].

Although epidemiologic data are limited, a hypothesized explanation for how undernutrition may protect against rotavirus diarrhoea is environmental enteric dysfunction [8]. Environmental enteric dysfunction (EED) is a complex intestinal disorder resulting from chronic exposure to enteric pathogens, and is ubiquitous among children living in areas with inadequate water, sanitation, and hygiene (WASH) [10–13]. Persistent exposure to enteric pathogens over time yields morphological changes in the gut such as blunting of intestinal villa, inflammation, epithelial damage, and reduced absorption of nutrients [11–14]. The synergism between malnutrition and EED yields a cyclical relationship in which malnourished children are more likely to develop EED and in turn remain malnourished due to inadequate absorption of nutrients in the gut and protein wasting [6, 11]. The high prevalence of malnutrition and EED in developing countries such as Zambia is also hypothesized to play a role in poor oral rotavirus vaccine performance in low and middle-income countries (LMICs) compared to higher income countries [15, 16].

While the morphological changes associated with EED may increase susceptibility to invasive enteric pathogens such as rotavirus and alter response to oral rotavirus vaccines, no studies to date have examined the impact of EED on the association between nutritional status and rotavirus diarrhoea. We aimed to address this critical research gap by examining the associations between nutritional status, EED, and rotavirus diarrhoea prevalence among Zambian children.

## Materials and methods

This study was approved by the University of Zambia Biomedical Research Ethics Committee, University of North Carolina at Chapel Hill Institutional Review Board and the Zambian Ministry of Health under Reference No. 014-09-11. All study participants provided written informed consent. Data in this analysis are from a case-control study evaluating the effectiveness of Rotarix™ vaccine against severe gastroenteritis in the Lusaka and Copperbelt provinces of Zambia in 2012–2013 already published elsewhere [17]. Case-patients were children that were identified in facility-based survey with moderate to severe gastroenteritis, were age-eligible to have received Rotarix vaccination, and had a stool specimen positive for rotavirus diarrhoea by enzyme immunoassay. Moderate to severe gastroenteritis was defined as having one or more of the following signs or symptoms: dehydration evidenced by sunken eyes, loss of

normal skin turgor, or requiring intravenous resuscitation; dysentery (diarrhoea with blood in stool); or hospitalization. Control-patients were either: 1) children with moderate to severe gastroenteritis identified in the facility-survey but were negative for rotavirus diarrhoea or, 2) children under the age of 5 who attended clinics presenting with non-diarrhoea-related complaints and have not yet produced stool that day.

## Sampling strategy and study population

For facility-based surveys, three Lusaka District facilities and one facility in each of the other 3 districts in Lusaka Province were selected as survey sites, as well as two Ndola District facilities from the Copperbelt Province that fell in or near the chosen population clusters in the community-based survey. Districts were selected to purposefully include a mixture of urban, peri-urban, and rural populations. Facilities with in-patient departments, sufficient patient volume, space to support study activities and geographic variation were prioritized for selection. Cases and controls identified in facility-based surveys were a systematic sample of outpatients and inpatients. Given the studies objective to evaluate vaccine effectiveness within a 1–2 year timeline, sampling was age-stratified to ensure that two-thirds of the sample was under the age of 2 years at the time of enrollment and therefore age-eligible for Rotarix vaccination. Patients with moderate to severe diarrhoea were sampled from facility inpatient departments, while patients asymptomatic for diarrhoea and presenting with non-diarrhoea related complaints were sampled from outpatient departments.

For this analysis, the sample was restricted to 711 children ($\leq$ 5 years old) and their parents/caregivers (15–17 years and mother of the child or $\geq$18 years old) after exclusion of 466 children with missing values for the exposure, outcome, or covariates of interest.

## Data collection

Parents/caregivers completed an interview-administered questionnaire which collected basic information about the child and parent/caregiver, household demographics, and child health status (e.g., occurrence of diarrhoeal disease, fever). Child vaccination history was verified using under-5 cards when available and, for a subset of children whose cards were not available, using health facility registers. Child weight, height, mid-arm circumference and skin fold measurements were measured by interviewers upon survey completion. Standard WHO collection procedures were used to collect approximately 10–15 mL of stool from all study participants. In addition to being tested for the presence of rotavirus, a subset of stool samples (~20%) were evaluated using commercial ELISA kits for the presence of faecal biomarkers of EED, namely: alpha-1 anti-trypsin (AAT), myeloperoxidase (MPO), and calprotectin (CALP).

## Outcomes

The primary outcome was rotavirus-specific diarrhoea, defined as diarrhoeal illness with a stool specimen positive for rotavirus using enzyme immunoassay (EIA).

## Primary exposure: Nutritional status

Nutritional status at the time of survey completion was measured using weight-for-age Z-score (WAZ), weight-for-height Z-Score (WHZ), and height-for-age Z score (HAZ) calculated according to WHO child growth standards [18]. Children with WAZ less than -2 were categorized as being malnourished, while children with WAZ greater than or equal to -2 were categorized as having "normal" nutritional status. Using the same convention, children with HAZ or WHZ less than -2 were categorized as being stunted and/or wasted, respectively

## Statistical analysis

The association between nutritional status and prevalence of rotavirus diarrhoeal disease (compared to no diarrhoeal disease) was examined using multivariate logistic regression. Chi-squared tests and t-tests were used to examine bivariate associations between the presence/absence of rotavirus diarrhoea and covariates specified *a priori* as potential confounders based on directed acyclic graph (DAG) based analysis, including: household water source, household sanitation type, child's sex, breastfeeding at the time of the interview, Rotarix vaccination status (status at least 14 days before illness onset among cases), parent/caregiver age, and household socioeconomic status. Self-reported type of sanitation facility used by each household was categorized as a binary variable indicative of whether the toilet facility adequately separates faecal matter from human contact consistent with categories established by World Health Organization/United Nations Children's Fund at the time of the study [19]. The self-reported main source of drinking water for members of the household was similarly categorized as a binary variable indicating whether the source of water was improved or unimproved consistent with 2012 WHO/UNICEF standard categories for improved/unimproved water sources [19]. Principle component analysis was used to construct an index of household socioeconomic status using the presence/absence of household possessions (i.e. television, radio, phone). Fully adjusted multivariate regression models included all covariates associated with the outcome of rotavirus diarrhoea ($p < 0.20$) in univariate analysis. Multiple imputation was used for covariates with >5% and <40% missing values [20].

In the subset of participants with data on faecal biomarkers of EED, t-tests were used to examine bivariate associations between biomarkers and the exposures and outcome of interest. Categories for each biomarker were defined using the distribution each biomarker with 0 being "$\leq 25^{th}$ percentile"; 1 "$>25^{th}$ to $<75^{th}$ percentiles"; and 2 "$\geq 75^{th}$ percentile". The EED severity score ranged from 0 to 10, with higher scores suggestive of higher EED severity. Finally, separate adjusted multivariate logistic regression models were used to examine the association between each indicator of nutritional status (i.e. malnutrition, wasting, stunting) and rotavirus diarrhoea with and without adjustment for EED severity score. Fully adjusted models also included the same covariates associated with the outcome of rotavirus diarrhoea ($p < 0.20$) in bivariate analysis in the larger study population. Given the limitations of multiple imputation with small sample sizes [21], a complete case analysis was done for all covariates in models including EED severity score. Statistical significance for all parameter estimates was evaluated at the alpha significance level of $P < 0.05$. All analyses were conducted with STATA 15 (College Station, Texas).

## Results

The study population included 711 children, of whom 53% were female and 47% were male (Table 1). The median age of children was 1.1 years (IQR: 0.7, 1.8), and at the time of survey completion 55.4% of study participants had received at least one dose of Rotarix vaccination. The majority of households had access to improved water sources (89.9%) but did not have access to improved sanitation facilities that adequately separated faeces from human contact (59.2%). Among children with rotavirus diarrhoeal disease, 78% had mild diarrhoea while 22% had moderate-to-severe diarrhoea.

### Nutritional status and rotavirus diarrhoea

The prevalence of malnutrition, wasting, and stunting in the study population was 22.3%, 17.4%, and 37.7%, respectively (Table 2). In both unadjusted and adjusted analyses, we found evidence suggestive of a protective association between malnutrition and odds of rotavirus

**Table 1. Sociodemographics of study participants, Zambia, 2012–2013.**

| Characteristic | Total (N = 711) | Number of Participants (% of total) | Number of Rotavirus Diarrhea Cases (% of row) |
|---|---|---|---|
| **Sex** | | | |
| Female | | 378 (53.2) | 316 (83.6) |
| Male | | 333 (46.8) | 272 (81.7) |
| **Child age in years, Median (IQR)** | 1.1 (0.7, 1.8) | | - |
| **Parent or caregiver age, Median (IQR)** | 25 (22–30) | | - |
| **Breastfeeding at time of interview** | | | |
| Yes | | 327 (46.0) | 321 (98.2) |
| No | | 121 (17.0) | 118 (97.5) |
| Missing | | 263 (37.0) | 149 (56.7) |
| **Socioeconomic status (quartile)** | | | |
| 1st (lowest) | | 114 (16.0) | 98 (86.0) |
| 2nd | | 119 (16.7) | 96 (80.7) |
| 3rd | | 127 (17.9) | 103 (81.) |
| 4th | | 123 (17.3) | 97 (78.9) |
| Missing | | 228 (32.1) | 194 (85.1) |
| **Household water** | | | |
| Unimproved | | 72 (10.1) | 57 (79.2) |
| Improved | | 639 (89.9) | 531 (83.1) |
| **Household sanitation** | | | |
| Unimproved | | 421 (59.2) | 412 (97.9) |
| Improved | | 147 (20.7) | 143 (97.3) |
| Missing | | 143 (20.1) | 33 (23.1) |
| **Rotarix vaccination** | | | |
| None | | 317 (44.6) | 260 (82.0) |
| Dose 1 or both | | 394 (55.4) | 328 (83.3) |

diarrhoea. In analyses adjusting for socioeconomic status, compared to children normal nutritional status, malnourished children had 0.66 times the odds of having rotavirus diarrhoea (95% CI: 0.42, 1.0; p = 0.07). We did not identify evidence of associations between the presence

**Table 2. Association between malnutrition (WAZ<-2), wasting (WHZ<-2), stunting (HAZ<-2) and rotavirus diarrhoea, Zambia 2012–2013.**

| Nutritional status | | N (col %) | Rotavirus Diarrhea (row %) | Unadjusted OR (95% CI) | p-value | Adjusted OR [A] (95% CI) | p-value |
|---|---|---|---|---|---|---|---|
| **Malnutrition (N = 665)** | | | | | | | |
| | Normal | 517 (77.7) | 434 (84.0) | 1.00 | - | 1.00 | - |
| | Malnourished | 148 (22.3) | 115 (77.7) | 0.67 (0.32, 1.0) | 0.08 | 0.66 (0.42, 1.0) | 0.07 |
| **Wasting (N = 662)** | | | | | | | |
| | Normal | 547 (82.6) | 451 (82.5) | 1.00 | - | 1.00 | - |
| | Wasted | 115 (17.4) | 96 (83.5) | 1.08 (0.63, 1.8) | 0.79 | 1.06 (0.62, 1.8) | 0.82 |
| **Stunting (N = 708)** | | | | | | | |
| | Normal | 441 (62.3) | 366 (83.0) | 1.00 | - | 1.00 | - |
| | Stunted | 267 (37.7) | 220 (82.4) | 0.96 (0.64, 1.43) | 0.84 | 0.93 (0.62, 1.4) | 0.71 |

[A] Adjusted for socioeconomic status.

*p<0.05, **p<0.01.

**Table 3. Bivariate associations between malnutrition (WAZ<-2), wasting (WHZ<-2), stunting (HAZ<-2), rotavirus diarrhoeal disease, and environmental enteric dysfunction (EED), Zambia 2012–2013.**

| | N | Mean EED Severity Score | p-value |
|---|---|---|---|
| **Malnutrition** | | | |
| Normal | 104 | 4.89 | 0.68 |
| Malnourished | 36 | 5.06 | |
| **Wasting** | | | |
| Normal | 121 | 4.78 | 0.04* |
| Wasted | 18 | 5.94 | |
| **Stunting** | | | |
| Normal | 85 | 5.28 | 0.02* |
| Stunted | 71 | 4.42 | |
| **Rotavirus Diarrhea** | | | |
| No diarrhea | 82 | 5.66 | 0.02* |
| Rotavirus-positive diarrhea | 31 | 4.68 | |

*p<0.05, **p<0.01.

of rotavirus diarrhoea and wasting (aOR = 1.06; 95% CI (0.62, 1.8); p = 0.82) or stunting (aOR = 0.93; 95% CI (0.62, 1.4); p = 0.71).

## Environmental enteric dysfunction (EED), nutritional status, and rotavirus diarrhoea

In the subset of children with available faecal markers of EED, malnourished children and wasted children had on average higher EED severity scores compared to children with normal nutritional status, but this difference was only statistically significant for wasting (p = 0.04; Table 3). Children who were stunted had on average lower EED severity scores compared to children with normal nutritional status (p = 0.02). Finally, average EED severity score was significantly higher among children asymptomatic for diarrhoeal disease compared to children with rotavirus-positive diarrhoea (p = 0.02). Odds of rotavirus diarrhoeal disease did not differ significantly based on nutritional status measured by malnutrition, wasting, or stunting in unadjusted or adjusted analyses (Table 4).

**Table 4. Association between malnutrition (WAZ<-2), wasting (WHZ<-2), stunting (HAZ<-2) and rotavirus diarrhoea in a subset of children with available measurements of environmental enteric dysfunction (EED), Zambia 2012–2013.**

| Nutritional status | | N (col %) | Rotavirus Diarrhea (row %) | Unadjusted OR (95% CI) | p-value | Adjusted OR [A] (95% CI) | p-value |
|---|---|---|---|---|---|---|---|
| **Malnutrition (N = 103)** | | | | | | | |
| | Normal | 77 (74.8) | 19 (24.7) | 1.00 | - | 1.00 | - |
| | Malnourished | 26 (25.2) | 8 (30.8) | 1.4 (0.51, 3.6) | 0.54 | 1.00 (0.34, 2.9) | 1.00 |
| **Wasting (N = 102)** | | | | | | | |
| | Normal | 87 (85.3) | 23 (26.4) | 1.00 | - | 1.00 | - |
| | Wasted | 15 (14.7) | 4 (26.7) | 1.01 (0.29, 3.5) | 0.99 | 0.77 (0.19, 3.1) | 0.71 |
| **Stunting (N = 108)** | | | | | | | |
| | Normal | 66 (61.1) | 16 (24.2) | 1.00 | - | 1.00 | - |
| | Stunted | 42 (38.9) | 15 (35.7) | 1.7 (0.75, 4.0) | 0.20 | 1.1 (0.40, 2.9) | 0.88 |

[A] Adjusted for socioeconomic status and environmental enteric dysfunction (EED) severity score.

*p<0.05 **p<0.01.

## Discussion

In this sample of Zambian children living in two of Zambia's 10 provinces, we identified a protective association between malnutrition and presence of diarrhoea due to rotavirus. In a subset of participants with available data on faecal biomarkers of EED, we found no evidence of associations between nutritional status and rotavirus diarrhoea but identified higher average EED severity scores among children with no diarrhoea compared to those with rotavirus positive diarrhoeal disease. We contribute to existing evidence suggestive of a protective association between malnutrition and rotavirus diarrhoea, and suggest that changes associated with EED may diminish the capability of rotavirus to invade gut epithelial cells.

Our findings of a protective association between malnutrition and rotavirus diarrhoea are consistent with existing observational studies examining the association between nutritional status and susceptibility to rotavirus infection without available data on EED [7–9]. The morphological changes associated with EED develop in response to chronic episodes of enteric infection in settings with inadequate WASH. In settings with persistent exposure to faecal pathogens, it is plausible that the pro-inflammatory and hyperactivated immune states associated with EED may develop as the body's defense to future pathogenic infiltration [11, 22]. Malnutrition and constant exposure to fecal contamination is also hypothesized to influence the composition of gut-microbiota, which may influence susceptibility to rotavirus [8, 11]. For example, in a study conducted by Kuss et al antibiotic depletion of gut microbiota in mice was associated with a reduced ability of polio viruses to infect host cells in the small intestine [23]. Although the effects of EED on microbial ecology are poorly understood, alterations in the composition of commensal bacteria may mediate the association between malnutrition and susceptibility to rotavirus infection.

Contrary to longitudinal research from Bangladesh, we did not find evidence of associations between stunting or wasting and odds of rotavirus diarrhoea even before adjustment for EED [8]. Risk factors for stunting are multifactorial and intergenerational, with maternal and intrauterine exposures acting as leading risk factors for growth faltering in early life and postnatal exposures such as frequent enteric infections becoming increasingly important as children get older [6, 24, 25]. Future research examining the association between nutritional status, EED and diarrhoeal disease should therefore utilize longitudinal follow-up that incorporates measurements of maternal and intrauterine predictors of childhood nutritional status over time.

While malnutrition and stunting occur due to long-term undernutrition, wasting is a result of acute significant food shortages and/or disease [26]. This secondary analysis, although based on data arising from a case-control design, utilizes a cross-sectional analysis to examine associations between nutritional status and prevalence of diarrhoeal disease at one-time point. Thus, a child's nutritional status as measured by wasting at the time of survey completion may not be indicative of their long-term nutritional status and therefore may not be likely to predict their odds of rotavirus diarrhoea. Although it is plausible that children with diarrhoeal disease would be more likely to be wasted at the time of survey completion, 78% of children with rotavirus diarrhoea had mild diarrhoea which may not have resulted in significant wasting at the anthropometric measurements were taken.

Our findings of a lack of an association between nutritional status and rotavirus diarrhoea in the subset of participants with available data on EED may indicate that there is no association between nutritional status and rotavirus diarrhoea, or alternatively that our limited sample size of participants with available data on EED markers did not provide us enough statistical power to identify existing associations. Average EED severity score was not significantly different between children who were malnourished compared to those that were not, while children who were stunted had lower average EED score compared to those that were not stunted. These findings are contrary to existing knowledge on the long term sequalae of

EED, which include malnutrition and stunting [27]. Notably, the EED severity index utilized for this analysis relied on three (AAT, MPO, CALP) of many existing biomarkers which have been shown to provide information associated with the morphological changes occurring in the gut due to EED [11, 13, 27]. For example, while AAT, MPO, and CALP provide information on intestinal permeability and inflammation [28–31], our summary EED severity score did not capture gut morphological changes such as enterocyte damage or intestinal absorption that would likely have implications for child nutritional status. The lack of clear diagnostic criteria for EED as well as the high prevalence of histological changes associated with EED among children living in settings with inadequate water and sanitation further complicate identification of the presence and extent of EED [10]. The high prevalence of EED among children in Zambia likely further limited our ability to demonstrate any existing differences in EED severity by nutritional status in our study population. Future studies utilizing larger sample sizes and a broader range of faecal and serum biomarkers indicative of morphological changes associated with EED as well as diagnostic methods such as endoscopy and biopsy are needed to confirm the findings of this analysis.

Evidence suggestive of EED as an explanatory factor for poor oral vaccine performance in developing countries are cohesive with the findings of this analysis [15]. We contribute to evidence suggesting that the same mechanisms that may prevent live attenuated viruses in oral vaccines from replicating in the guts of children with EED may also protect them from wild rotavirus infection [8]. The association between EED, oral vaccine performance, and diarrhoeal disease remains poorly understood and may vary by specific enteric pathogens (e.g. invasive vs. non-invasive, etc.). Longitudinal research among the growing population of infants and children who receive oral vaccines in developing countries is critically needed to better understand the causal mechanisms by which better-nourished children may be more susceptible to rotavirus infection in early life.

This analysis has several limitations. In addition to our limited sample size, we utilized a cross-sectional data and therefore cannot make any causal claims regarding the association between nutritional status, EED, and rotavirus diarrhoea. Data-driven selection of covariates to include in adjusted models based on univariate associations with the outcome of diarrhoeal illness may have increased susceptibility for unmeasured confounding. Finally, our use of multiple imputation in the overall study population and complete case analysis in the subset of participants with available data on EED may have biased our estimates if socioeconomic status was not missing at random in our study population [20].

## Conclusion

We provide the first estimates, to our knowledge, of the association between nutritional status and prevalence of rotavirus diarrhoea after adjustment for EED. In a sample of children under the age of 5 in two provinces in Zambia, before adjustment for EED malnourished children had a significantly lower odds of rotavirus diarrhoea compared to children with normal nutritional status. After adjustment for EED, we found no associations between nutritional status and rotavirus diarrhoea. Our findings not only identify a high prevalence of malnutrition, wasting, and stunting among children in Zambia, but underscore the urgent need for the inclusion of measures of EED in future research studies needed to better understand complex interactions between nutritional status, diarrhoeal disease, and live oral vaccine uptake.

## Acknowledgments

The authors thank the participants and the CIDRZ staff on the ACADEMIC study for their important contributions. The authors also thank the Biomedical Science students of the

University of Zambia, College of Health Sciences (Malumbe Michelo, Lunenge Kasukumya and Diana Zulu) for their help with the experimental work.

## Author Contributions

**Conceptualization:** Michelo Simuyandi, Samuel Bosomprah, Roma Chilengi.

**Formal analysis:** Aybüke Koyuncu.

**Methodology:** Roma Chilengi.

**Writing – original draft:** Aybüke Koyuncu.

**Writing – review & editing:** Samuel Bosomprah, Roma Chilengi.

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
