## [Decision Letter · Decision Letter 0]

31 Jul 2020

PONE-D-20-08984

Nutritional Status, Environmental Enteric Dysfunction, and Prevalence of Rotavirus Diarrhoea Among Children in Zambia

PLOS ONE

Dear Dr. Koyuncu,

Thank you for submitting your manuscript to PLOS ONE. After careful consideration, we feel that it has merit but does not fully meet PLOS ONE’s publication criteria as it currently stands. Therefore, we invite you to submit a revised version of the manuscript that addresses the points raised during the review process.

We look forward to receiving your revised manuscript.

Kind regards,

Pradeep Dudeja

Academic Editor

PLOS ONE

Journal Requirements:

2.We note that you have indicated that data from this study are available upon request. PLOS only allows data to be available upon request if there are legal or ethical restrictions on sharing data publicly. For information on unacceptable data access restrictions, please see http://journals.plos.org/plosone/s/data-availability#loc-unacceptable-data-access-restrictions.

3.Thank you for stating in your Funding Statement:

 [This work was undertaken with partial support from the Bill and Melinda Gates Foundation (https://www.gatesfoundation.org/)  grant number OPP1033211. Funding from the Bill and Melinda Gates Foundation was awarded to Jeffrey Stringer, who is not a part of the authorship team for this manuscript. The funders had no role in study design, data collection and analysis, decision to publish, or preparation of the manuscript.]. 

Reviewers' comments:

Reviewer's Responses to Questions

**Comments to the Author**

1. Is the manuscript technically sound, and do the data support the conclusions?

Reviewer #1: Yes

2. Has the statistical analysis been performed appropriately and rigorously? 

Reviewer #1: Yes

3. Have the authors made all data underlying the findings in their manuscript fully available?

Reviewer #1: Yes

4. Is the manuscript presented in an intelligible fashion and written in standard English?

Reviewer #1: Yes

5. Review Comments to the Author

Reviewer #1: Koyuncu et al. expand on a study performed from 2012-2013 in children in Zambia to determine whether environmental enteric dysfunction impacts rotavirus infection. The authors report that morphological changes that likely result from environmental enteric dysfunction may paradoxically provide protection against rotavirus infection in children. The manuscript is thought provoking, well written and provides novel findings. The manuscript could benefit from more discussion on potential scientific mechanisms that may provide the protection, I feel that this would strengthen the manuscripts findings. Overall, the authors’ background, methods, findings and conlcusions are logical, well laid out and compelling.

Minor Comments:

1. Please define LMIC (line 74)

2. I think table 1 would be easier to read if the characteristics were bolded or underlined to separate out the characteristics ie: bold or underline sex and then male female would be easier to identify as separate from the next characteristic Child age in years. A structure similar to table 2 with characteristics separated with lines would also work well.

3. Line 202 different should read differ.

4. Please elaborate in the discussion on why morphological changes in the gut and inflammation may impact the capability of rotavirus to invade the gastrointestinal tract. Are there any scientific papers that suggest that villous blunting diminishes rotavirus infection or whether increased cell shedding or altered microbiota in malnourished children contributes to decreased rotavirus impact? If so, these studies would be helpful to discuss and reference and would provide additional support for the authors’ conclusions.

6. PLOS authors have the option to publish the peer review history of their article (what does this mean?). If published, this will include your full peer review and any attached files.

Reviewer #1: No

---

## [Author Response · Author response to Decision Letter 0]

14 Sep 2020

Responses to editor comments:

Thank you. We have updated our manuscript to PLOS ONE's style requirements and included the requested clarifications in our revised cover letter. 

Responses to reviewer comments:

1. Please define LMIC (line 74)

Thank you for noting this, the correction has been made.

2. I think table 1 would be easier to read if the characteristics were bolded or underlined to separate out the characteristics ie: bold or underline sex and then male female would be easier to identify as separate from the next characteristic Child age in years. A structure similar to table 2 with characteristics separated with lines would also work well.

Thank you for this suggestion, we have updated table 1 accordingly.

3. Line 202 different should read differ.

Thank you, the correction has been made.

4. Please elaborate in the discussion on why morphological changes in the gut and inflammation may impact the capability of rotavirus to invade the gastrointestinal tract. Are there any scientific papers that suggest that villous blunting diminishes rotavirus infection or whether increased cell shedding or altered microbiota in malnourished children contributes to decreased rotavirus impact? If so, these studies would be helpful to discuss and reference and would provide additional support for the authors’ conclusions.

Thank you for the opportunity to elaborate. The discussion section has been amended to include the following:

“The morphological changes associated with EED develop in response to chronic episodes of enteric infection in settings with inadequate WASH. In settings with persistent exposure to faecal pathogens, it is plausible that the pro-inflammatory and hyperactivated immune states associated with EED may develop as the body’s defense to future pathogenic infiltration [11,22]. Malnutrition and constant exposure to fecal contamination is also hypothesized to influence the composition of gut-microbiota, which may influence susceptibility to rotavirus [8,11]. For example, in a study conducted by Kuss et al antibiotic depletion of gut microbiota in mice was associated with a reduced ability of polio viruses to infect host cells in the small intestine [23]. Although the effects of EED on microbial ecology are poorly understood, alterations in the composition of commensal bacteria may mediate the association between malnutrition and susceptibility to rotavirus infection.”

---

## [Editor Report · Decision Letter 1]

23 Sep 2020

Nutritional status, environmental enteric dysfunction, and prevalence of rotavirus diarrhoea among children in Zambia

PONE-D-20-08984R1

Dear Dr. Koyuncu,

We’re pleased to inform you that your manuscript has been judged scientifically suitable for publication and will be formally accepted for publication once it meets all outstanding technical requirements.

Kind regards,

Pradeep Dudeja

Academic Editor

PLOS ONE
---

## [Editor Report · Acceptance letter]

25 Sep 2020

PONE-D-20-08984R1 

Nutritional status, environmental enteric dysfunction, and prevalence of rotavirus diarrhoea among children in Zambia 

Dear Dr. Koyuncu:

I'm pleased to inform you that your manuscript has been deemed suitable for publication in PLOS ONE. Congratulations! Your manuscript is now with our production department. 

Kind regards, 

on behalf of

Dr. Pradeep Dudeja 

Academic Editor

PLOS ONE